# Paper-Based Multiplex Sensors for the Optical Detection of Plant Stress

**DOI:** 10.3390/mi14020314

**Published:** 2023-01-26

**Authors:** Marie Zedler, Sze Wai Tse, Antonio Ruiz-Gonzalez, Jim Haseloff

**Affiliations:** Department of Plant Sciences, University of Cambridge, Downing St., Cambridge CB2 3EA, UK

**Keywords:** optoelectronic nose, plant stress, sensors, smart agriculture, low-cost

## Abstract

The rising population and the ongoing climate crisis call for improved means to monitor and optimise agriculture. A promising approach to tackle current challenges in food production is the early diagnosis of plant diseases through non-invasive methods, such as the detection of volatiles. However, current devices for detection of multiple volatiles are based on electronic noses, which are expensive, require complex circuit assembly, may involve metal oxides with heating elements, and cannot easily be adapted for some applications that require miniaturisation or limit front-end use of electronic components. To address these challenges, a low-cost optoelectronic nose using chemo-responsive colorimetric dyes drop-casted onto filter paper has been developed in the current work. The final sensors could be used for the quantitative detection of up to six plant volatiles through changes in colour intensities with a sub-ppm level limit of detection, one of the lowest limits of detection reported so far using colorimetric gas sensors. Sensor colouration could be analysed using a low-cost spectrometer and the results could be processed using a microcontroller. The measured volatiles could be used for the early detection of plant abiotic stress as early as two days after exposure to two different stresses: high salinity and starvation. This approach allowed a lowering of costs to GBP 1 per diagnostic sensing paper. Furthermore, the small size of the paper sensors allows for their use in confined settings, such as Petri dishes. This detection of abiotic stress could be easily achieved by exposing the devices to living plants for 1 h. This technology has the potential to be used for monitoring of plant development in field applications, early recognition of stress, implementation of preventative measures, and mitigation of harvest losses.

## 1. Introduction

The world population is projected to grow to around 10 billion in 2050. This leaves us with a future food mi, demanding an increase in food production of up to 56% by 2050 [1]. However, environmental stresses such as extreme weather events or agricultural malpractices, including fertiliser overuse and soil tilling, cause significant losses in agriculture [2]. In particular, soil degradation is estimated to cause economic losses that reach GBP 1.2 billion per year in the UK only [3]. As such, a rising population as well as the changing climate require new solutions for agriculture that offer growth in a sustainable way. By detecting the physiological stress of plants early on, production losses can be minimised through the implementation of preventative measures against the stress source [4]. To date, several technologies that detect plant stressors have been developed. Some examples of non-invasive devices for analysing plant health include proximal optical sensors to monitor nitrogen, imaging methods to detect plant diseases, as well as smartphone-based methods [5]. 

Recently, some imaging approaches have been reported for the determination of H_2_O_2_ in plants using implanted carbon nanotubes and measuring their infrared absorption [6]. However, this approach is complex, since it requires plant leaf infiltration with carbon nanotubes. Moreover, although a release of hydrogen peroxide has been observed in plants subjected to biotic and abiotic stress, this biomarker is not selective towards the source of stress [7], and its detection requires expensive equipment for the measurement of infrared (IR) reflectance, limiting the applicability of this technology within real-world settings. Recent studies have also explored electrochemical methods, employing printed electronics and flexible materials such as silver/reduced graphene nanocomposites for the determination of toxic gases [8]. Although electrochemical-based methods are less expensive compared with traditional optical approaches and they reduce the needs for expensive equipment, the sensing performance, in terms of sensitivity and selectivity, tends to be relatively low [8]. Li et al. [9] developed a real-time electrochemical sensor for the detection of leaf volatiles using a carbon nanotube (CNT)/graphite composite. The final device could quantify the concentrations of organic volatiles, such as methyl jasmonate, within 20 s. However, the device showed significant interference from other common metabolites, such as 1-hexenal and 2-phenylethanol, increasing the complexity of the data analysis, and the limit of detection was relatively high in the range of 5 ppm.

Another approach to monitor plant health that is becoming increasingly popular is remote sensing for the detection or prediction of plant diseases in crop fields [10]. Typically, unmanned aerial vehicles (UAVs) are used to facilitate aerial imaging with cameras and sensors. These include the less expensive red, green, blue (RGB) detectors, multi- or hyper-spectral cameras, or thermal cameras capable of detecting stress-induced changes in plant metabolism. These technologies are becoming cheaper and more sophisticated, but are still far from affordable, in particular for small-scale farmers. Background noise and varying field conditions or symptom phenotypes present further challenges to application of these sensors. In addition, some applicability of electrophysiological monitoring for water stress could be demonstrated [11]. However, most of these techniques using optical mechanisms cannot detect external biotic and abiotic stresses of individual plants. Moreover, there is often a lack of high temporal resolution, hindering the prompt and effective reaction to plant stresses. 

The limitations observed in electrochemical and optical sensors could potentially be addressed by the combination of multiple low-cost devices in the form of electronic noses. Electronic noses, or e-noses, are devices capable of sensing odours by mimicking the human olfaction system. These systems incorporate multiple sensors with varying selectivities and sensitivities, which can be used to profile complex gas mixtures. In this case, gases react with each sensor contained in the e-nose to trigger a distinct signal change. In typical metal-oxide based sensors, the increase in gas concentrations leads to a change in electrical resistance, which can be determined by the output voltage. However, electronic noses often require complex circuit assembly, involving metal oxides with heating elements, and printed circuit boards, and often, they cannot be easily modified for the tailored detection of new biomarkers. As such, despite the wide range of commercially available sensing devices, the flexibility of such systems to be adapted for specific applications such as plant stress monitoring is limited. Furthermore, complex manufacturing processes lead to the relatively high costs of these devices, in the range of USD 200 [12], and they generally require specialised personnel to assemble the devices and analyse the results. 

The human olfactory system can discriminate trillions of odours with only hundreds of olfactory receptors by representing them as a combinatorial code [13]. An emergent technology related to e-noses was developed by Suslick et al. [14] and provides an alternative to current devices. These are optoelectronic noses that produce a unique pattern of colorimetric changes in response to a given analyte (Figure 1).

Optoelectronic noses make use of multiple chemically responsive dyes such as metalloproteins or Reichardt reagents that induce a colour change upon their exposure to certain volatiles. These dyes can be further modified through the incorporation of reactive chemical species, such as silver nitrate, to change the mechanism of action and selectivity [15]. Consequently, exposure to a particular volatile leads to a set of colour changes on the multiplexed sensor, and these can subsequently be analysed and classified as a specific colour fingerprint. This method results in high sensitivity and good discrimination between similar analytes. Moreover, it offers a low-cost and simple solution to the detection of a wide range of compounds. Thus far, applications include the detection of pesticides, explosives, or the early diagnosis of lung cancer [16,17]. However, their application in the prevention of crop losses has been limited. Recently, Li et al. [5] developed an optoelectronic nose based on gold nanoparticles for the detection of Phytophora infestans infections on tomato plants. This device could diagnose tomato leaves within 1 day of inoculation by quantifying the amounts of 2-hexen-1-al. Although this device showed a high sensitivity, it required functionalised thin gold nanorods (around 20 nm), increasing the price of the final setup, and the sensors were only tested on individual leaves instead of whole plants. 

As mentioned, one of the key limitations of optoelectronic noses is the high cost, resulting from the use of sometimes expensive dyes and spectrophotometric meters for analysing the results [18]. This work reports a low-cost optoelectronic nose as an alternative to current approaches in the study of plant responses to abiotic stress. These sensors can detect a variety of specific gas analytes based on colorimetric changes caused by interactions between chromophores in a dye with the analyte. The devices can be easily fabricated by drop casting the selective dyes mixtures onto filter paper. We focused on the detection of volatile organic compounds (VOC), such as 2-hexenal, which are known to be released from soil or plant leaves under certain conditions of biotic or abiotic stress [19]. Initially, a systematic study on how different reactive dyes can be combined to enhance the sensitivity towards certain analytes was conducted. Common chromophores, including methyl red or Reichardt’s dye, were deposited onto porous filter paper using different solvents. The impact of the solvent on the sensing performance of the sensors was determined. Moreover, the selectivity of the sensor was enhanced by combination of the solution with molecular pores, such as cyclodextrin, and optically active materials, such as graphene quantum dots. Finally, the VOC profiles of *Marchantia polymorpha*, a model liverwort plant widely used in research, were screened to inform us about their health status. Two different abiotic stress sources were tested, i.e., nutrient deficiency and high salinity, and the changes in volatile profiles in each case were determined. The final devices were able to determine the stress response within 1 day of exposure to each stress source, and the price of each device was as low as GBP 1. Finally, to allow a standardisation of measurements, we developed a spectrometer case and incorporated a live display of colour data. 

## 2. Materials and Methods

### 2.1. Materials

All compounds and reagents were purchased from Sigma Aldrich unless otherwise specified; silver nitrate, cyclodextrin, methyl red, acetone, ethanol, 2-hexen-1-al, acetic acid, blue graphene quantum dots, bromocresol green, phenyl red, Reichardt’s dye, nickel(III) phthalocyanine tetrasulfonic acid tetrasodium salt, and MnTBAP chloride. The Gamborg’s growth medium used was Gamborg’s B-5 Basal Medium with minimal organics, and was purchased from Sigma-Aldrich. Filter paper was purchased from Amersham Protran. Components for the spectrometer include a rotor (Small Reduction Stepper Motor - 5VDC 32-Step 1/16 Gearing, from The Pi Hut), an LED ring (16-pixel RGB LED ring Rainbow FC-102 for Arduino), and the Wio Terminal (from Seeed Studio). 

### 2.2. Fabrication of Sensors and Deposition of Dyes

Solutions containing multiple colorimetric dyes with proven performance for gas detection, as well as combinations of dyes with different solvents and matrices, were tested. In each case, the dye solutions were prepared by mixing 5 mM of each dye in 1.5 mL Eppendorf tubes. To lower manufacturing costs, disposable sensors were produced by deposition of multiple dyes onto filter paper by drop casting. Filter paper was cut into circular pieces with a diameter of about 5.5 cm and the space was divided into 8 sections of equal size. On each of the 8 sections, 20 µL of dye was drop-casted, which formed a coloured spot of about 1–2 cm diameter. The manufacturing process of these sensors was fast and took less than one minute per sensor paper. To ensure low interference from solvent residues, each multiplexed sensing paper was left to dry for at least 4 h. A list of reactive dye combinations used in this study is provided in Figure 2. These dyes were combined with different solvents, as well as analyte-binding compounds such as cyclodextrin (CD), to tune the selectivity of the reactions. 

### 2.3. Testing of Sensitive Dye Response towards Different Gases

Initially, to demonstrate the selective reactivity of the sensing dyes in the presence of different volatiles, a series of tests were conducted with the following gases: water vapour, ethanol, acetic acid, and acetone. This experiment allowed us to determine potential interference due to water in our experiments, since it is ubiquitously present in all environments, and the changes in sensitivity of each dye towards different chemical functionalities. Moreover, the chosen organic volatiles (acetone, acetic acid, and ethanol) allowed us to study the impact of different functional groups on the response of the final devices. In each case, the sensing papers were subjected to 10 ppm of each gas inside a sealed pyrex borosilicate bottle, and the changes in colour were determined by a low-cost TCS34725 (Adafruit)-based spectrometer after 2 h.

After conducting this initial test using simple organic volatiles and water, the sensors were tested in the presence of trans-2-hexen-1-al, indole-3-acetic acid (auxin), and tryptophol. These molecules are involved in the growth and stress response of plants. In particular, trans-2-hexen-1-al is a volatile organic compound (VOC) emitted by wounded or stressed plants, indole-3-acetic acid is the most common naturally occurring phytohormone of the auxin class, and tryptophol is a quorum sensing molecule (QSM) found during fungal infections. We tested the sensitivity of the dyes in the presence of 10 ppm of each gas separately. The exact values were calculated using the following formula (1): (1)$$c=\frac{22.4\times p\times d\times V_{1}}{M\times V_{2}}$$
where *c* denotes concentration in ppm, *p* denotes purity, *d* denotes density in g/mL, *V_1_* denotes the volume injected in µL, M denotes molecular weight, and *V_2_* denotes the volume in the bottle in L. The colorimetric sensor arrays were left in sealed 1 L glass bottles for about 2 h to react with the gases. Color changes in the drop-casted dyes were then determined. A simple and low-cost RGB colour sensor (Adafruit TCS34725) was used, connected to an Arduino board for data collection. This sensor component also had an in-built white LED next to its RGB sensor. The TCS34725 contained a 3 × 4 photodiode array, consisting of red-filtered, green-filtered, blue-filtered, and clear (unfiltered) photodiodes. This RGB sensor quantifies the level of reflected light from the white LED source and measures the following wavelengths: 465 nm (blue), 525 nm (green), and 615 nm (red). 

Connections to the colour sensor were soldered to allow for stable electrical contacts and to hold the components firmly in place. During the measurement, a paper sensor was placed onto the colour sensor, so that the LED of the colour sensor evenly illuminated the sample. RGB values for each coloured spot on the filter paper were recorded. The measurements were recorded using the microcontroller Wio Terminal, which could potentially be used to automate the whole data extraction and calculation process. 

### 2.4. Dye Screening

A total of 19 dye mixtures were tested in this work. The composition of each mixture can be found in Figure 2. To select the most successful dyes, the data analysis method proposed by Li et al. was adopted [5]. In this case, the Euclidean distance of all colour measurements (each dye separately) from the control was calculated. These were compared to control sensing papers that had not been exposed to any of the gases. The Euclidean distance was determined as the distance between two points in three-dimensional colour space. Given two points, where R = *x*, G = *y*, and B = *z* (test RGB values = (*x*_1_, *y*_1_, *z*_1_) and control RGB values = (*x*_0_, *y*_0_, *z*_0_)), the distance (*d*) between them could be calculated as (2): *d* = p(*x*_1_
*− x*_0_)^2^ + (*y*_1_
*− y*_0_)^2^ + (*z*_1_
*− z*_0_)^2^(2)

To screen for dyes that were most responsive towards each analyte, the average response of the sensors towards all tested gases was first calculated. Individual gases measurements were then subtracted from this average and plotted. This graph allowed us to select dye combinations that produced higher absolute signals after exposure to each analyte. Dyes which showed the largest colour changes were chosen to maximise sensitivity. The three dyes with the most sensitive response to trans-2-hexen-1-al were selected, as well those with the best performance for each of the other gases except for water vapour. The final sensing papers used in the biological studies comprised a total of 8 dyes. Water vapour-sensitive dyes were excluded to avoid unnecessary cross-talk, since it is present at varying levels in most tested environments, and alone provides little information about plant stress responses. 

The optimised sensor papers were manufactured and tested in triplicate by exposure to each of the gases (acetic acid, acetone, ethanol, 2-hexen-1-al, indole-3-acetic acid, and tryptophol). After each experiment, 8 different Euclidean distance measurements were obtained. Given the high dimensionality of the data obtained, different gas volatile profiles could not be easily classified. To study the capability of our device to differentiate the studied gases, principal component analysis (PCA) was used. The analysis software BioVinci 2.0 was employed, allowing a reduction in data into two dimensions for the visualisation of plant subgroups. Moreover, the response of the sensing strips subjected to different concentrations of trans-2-hexen-1-al was tested. This allowed the determination of the detection limit of our sensors, and to also potentially quantify the gas concentrations over time during the in vivo experiments. In this case, the sensors were tested by exposure to analyte concentrations of 0.5, 1, 2.5, 5, 10, and 25 ppm. Similar to the previous test, the sensors were left for about 2 h to react with the gas in each case. 

### 2.5. Testing In Vivo Using Plants

Finally, the performance of paper sensors was tested in vivo using living plants. *Marchantia polymorpha* was chosen as the model plant, since they are easy and fast to grow, as well as to allow for simple testing procedures. Wild-type *Marchantia polymorpha* accessions Cam-1 (male) and Cam-2 (female) were used in the experiments. Gemmae and thallus cuttings of plants were grown and maintained in 0.5× Gamborg’s media (Gamborg’s B5 medium plus vitamins, Duchefa Biochemie G0210, pH 5.8) and 1.2% (*w*/*v*) agar (Melford capsules, A20021) at a continuous temperature of 21 °C under continuous light (light intensity of 150 μmol/m²/s). Plates containing 100 μg/mL of cefotaxime were used in the earliest two batches of tissue culture to prevent bacterial and fungal contamination.

Wild-type gemmae, with an age of about two weeks, were used in each case. The plants were tested in three different growth conditions: control, high salinity, and nutrient starvation. For the control measurements, Gamborg’s medium was used, which is a standard medium for in vitro culture of plants and consists of a nutrient blend of inorganic salts, vitamins, and carbohydrates. To induce salt stress, 50 mM NaCl was added to the standard GB medium. This concentration is high enough to stress the plant but is not lethal [20]. For the starvation medium, the standard GB medium was diluted to 1/100 of its original concentration in RO water. For each condition, triplicates of the plants and the respective growth medium were made. For test measurements of volatiles produced in plants, sensing strips were attached to the inner side of Petri dish lids and closed for at least 1 h. The sensors were then removed and the colorimetric changes induced by their exposure to the plant volatiles were measured using the TCS34725-based spectrometer. The experiment was run over the course of 5 days to study the dynamical changes in volatile profiles. Further, pictures of the plants were also taken each day to compare the visual differences between the growth conditions. 

### 2.6. Measuring Stress in Plants Using Electronic Noses for Comparison

For direct comparison of our results with electronic noses, in vivo sensing experiments were conducted using a combination of commercially available gas sensors with the *Marchantia polymorpha* model. A custom-made e-nose was developed containing MQ4, MQ7, and MQ131 (metal oxide sensors); Bosch BME680 (which contains multiple sensors to measure humidity, temperature, pressure, and total VOCs); and Sensiron SCD30 (a near infrared spectrometer-based sensor) to measure CO_2_. Concentrations of the target gases were calculated according to the manufacturer’s calibration. Sensors were pre-conditioned for 1 h prior the experiments in open air, or until a stable signal was reached. The sensors were connected to an Arduino UNO microcontroller board and assembled inside a plastic box, where tubes were connected through a plastic septum to avoid gas leaks. Moreover, to allow a direct quantification in Petri dishes, as conducted in the case of optoelectronic noses, the Petri dishes were modified to allow the connection of an air pump to drive the gas towards the electronic nose (Figure 3). A tube was also incorporated to connect the Petri dish with the electronic nose. Finally, Petri dishes containing growing Marchantia plants were sealed, and measurements were taken for at least 1 h, as in the case of the optoelectronic noses. 

### 2.7. Development of Low-Cost Spectrometer for Standardised Measurements

To facilitate regular measurements of the colorimetric sensing paper, a portable spectrometer was fabricated, comprising a 3D-printed case with sliding lids, a rotor to move the dye samples, and a display to show live colour results. The case enabled the establishment of standardised conditions for measurement, including background light intensity, since the sample needs to be in darkness during each measurement. Moreover, sliding lids were designed to allow for an easy exchange of samples (Figure 4). 

The 3D model of the case was obtained using the free modelling software SketchUp, and printed using an Ultimaker S3 printer with PLA filament. The case contained a holder for a rotor (Small Reduction Stepper Motor - 5VDC 32- Step 1/16 Gearing, from The Pi Hut) and a circular LED (16-pixel RGB LED ring Rainbow FC-102 for Arduino) to indicate the measurement status. A rotating disc plate to support the sensor papers was also printed. The dimensions of the disc fit the size of the measurement sensor paper, with a diameter of 5.5 cm. The rotor was programmed such that the disc rotated to one of the 8 evenly spread samples, 10 measurements were taken by the colour sensor, and then this process was repeated until all 8 samples were measured. Then, another paper sensor can be placed into the device for further measurements. A ring-shaped display of multiple RGB LEDs was used to indicate the measurement status of the spectrometer. For better user-friendliness and a live analysis of results, a Wio terminal with a 320x240 pixels LCD display was programmed to show sensor data in real time. The Wio terminal shows RGB and HEX values from the colour sensor as numerical values as well as RGB values in a graph. This device can potentially send data wirelessly to an online server for IoT applications. 

## 3. Results and Discussion

### 3.1. Fabrication and Screening of Chromogenic Dyes

The first step in fabrication of the sensing devices consisted of the selection of a combination of suitable dyes and sensing compositions. Optoelectronic noses typically involve chromogenic indicators, able to change the emitted light within the visible range [18]. In this case, four different chromogenic dyes were employed with a proven performance in volatile detection: bromocresol green (BG), methyl red (MR), bromophenol red (BR), and Reichardt’s dye (RD). These dyes could produce a measurable change within the visible spectrum upon exposure to organic volatiles that could be determined using a low-cost spectrometer. In all cases, red, blue, and green signals (corresponding to wavelengths 575, 535, and 445 nm, respectively) were measured using our low-cost TCS34725 spectrometer device. The responses towards multiple volatiles, including acetone, acetic acid, ethanol, water, 2-hexen-1-al, tryptophol, and auxins, were then determined. The results from the initial screening are shown in Figure 5a. In this case, the differences between the three-dimensional Euclidean distance of the control, not exposed to the target gas, and the one achieved on exposure to each gas was calculated. This value of Euclidean distance difference was used to determine the sensitivity of the dyes, since it provided information about the total magnitude change in colouration. 

To provide a comparative measurement of the sensitivities and to allow a study of the selectivity of each individual dye composition towards different volatiles, the average sensitivity achieved during the screening was calculated. This average sensitivity included the values achieved for each volatile studied in this work, i.e., water, ethanol, acetone, acetic acid, auxin, tryptophol, and 2-hexen-1-al. In each case, the sensitivity value was subtracted from the average measurement obtained across all the tests conducted on a single sensor. This measurement allowed us to determine those dye compositions with the highest absolute sensitivity towards each gas. Dyes with the highest difference in sensitivity compared to the average value when measured against specific volatiles were considered as optimal, since they showed the highest sensitivity, and the effect of interference due to other gases was lower. The impact of each dye composition on the final sensitivity is described in subsequent sections. 

Moreover, the correlation matrix with all the obtained results of each dye was calculated (Figure 5b). The final gas identification was conducted by principle component analysis of the final eight dye compositions. However, in some cases, such as RD + Water and BR + Water compositions, a correlation coefficient close to one was obtained, which indicated a potentially similar response of the dyes. The incorporation of multiple dyes with similar response profiles could potentially provide lower information of gas volatiles. As such, dyes with a high sensitivity towards target analytes, but low correlation with other dye compositions, were selected to help identify gases.

The sensors were selectively exposed to acetone, ethanol, and acetic acid volatiles to better study the impact of particular functional groups on dye metachromasia. These volatiles offer a similar molecular structure, where covalently bound oxygen is present in different configurations as ketone, alcohol, or carboxyl groups (Figure 5a). This study demonstrated clear selective colour changes among different dye sets on the sensor papers. Furthermore, candidate dye mixes were exposed to water vapour to determine responses to humidity. Different levels of water are present in plant culture vessels and could lead to inaccuracies in the detection. Therefore, sensors with a low response to water vapour were chosen.

#### 3.1.1. Polar Volatile Detection with Chromogenic and Solvatochromic Dyes

BG, MR, and BR were used as pH indicators due to changes in colour due to proton exchange in acid or basic environments. On the contrary, RD is solvatochromic, changing its colour as a consequence of the polarity of the solvents. As such, both groups of dyes provide useful characteristics for the development of the optoelectronic nose, given their strong response to environmental exposure to gases that leads to a change in H^+^ or polarity changes around the sensing strips. In particular, pH indicators are low-cost, providing the opportunity for developing accessible devices at a low cost. However, these dyes are not individually selective. Consequently, they were combined with other reactive species to confer specificity towards different gases. In all cases, the dyes were deposited by drop-casting onto a filter paper. This method reduced the manufacturing costs and led to a good signal reproducibility.

As part of the screening process, individual dyes as well as dyes combined with other reactive compounds were used. The sensors were exposed to simple analytes, including water, acetone, ethanol, and acetic acid, and plant health biomarkers, including auxin, tryptophol, and trans-2-hexen-1-ol. One of these compounds used to enhance selectivity was tetramethyl ammonium (TMA). TMA is a positively charged quaternary amine which can attract polar molecules given its charge. This compound was used with the pH indicators (Figure 5b). In the case of MR, its selectivity towards highly polar compounds, such as water and acetic acid, increased as expected. The signals, measured as the total Euclidean distance of the intensity measured from RGB measurements for water and acetic acid, were 150% and 187% higher than the average, respectively. However, this increase in sensitivity was lower in the case of BR and BG.

#### 3.1.2. Testing of Porphyrins as Additional Sensing Compounds

The combination of chromogenic and solvatochromic dyes provided a good sensitivity towards the target biomarker. These dyes could be easily combined with other compounds to enhance the selectivity of detection. Additional dyes have been tested in the literature, based on porphyrin dyes [21,22]. These dyes can form complexes with metal ions and interact selectively with other molecules due to their chemical properties. The interaction of the volatiles with porphyrins can promote a colour change due to the effects of the coordination of complexes with the coordinated ions. This interaction can be specific depending on the chemical structure of the porphyrins and the ion employed. As such, it has been widely employed as a functional compound in optoelectronic noses. Using these compounds, Palasuek et al. [22] developed an optoelectronic nose for the analysis of food samples. However, the price of these chemicals tends to be higher than those found for pH-sensitive dyes, hindering application in low-resource settings. 

Two different porphyrin derivatives were tested in the present study: nickel(III) phthalocyanine-tetrasulfonic acid tetrasodium salt and manganese (III) tetrakis (4-benzoic acid) porphyrin chloride (MnTBAP chloride). In the case of nickel(III) phthalocyanine, a low selectivity was observed, with no significantly higher sensitivity towards any of the studied chemicals. The average change rate for each volatile was, however, higher in general terms than the one observed in chromogenic dyes, indicating a high reactivity but low selectivity of this dye. On the contrary, MnTBAP showed a high sensitivity towards acetone, compared to the rest of volatiles. This sensitivity was one of the highest recorded during the screening and was 199.4% higher than the average measured sensitivity. As such, it could be selected as an acetone-specific sensitive dye.

#### 3.1.3. Cyclodextrin as an Adjuvant for Detection of Indole Biomarkers

Although the use of TMA enhanced the specificity of detection of polar volatiles, the sensitivity towards other metabolites of interest in the characterisation of plant abiotic stress was relatively low. In particular, these sensors did not raise a significantly high signal to allow the detection of indole molecules. Indoles are aromatic heterocyclic compounds that are typically derived from the amino acid tryptophan. Auxins, for example, are mostly synthesised from tryptophan and are a major player in plant growth and development [23]. This compound results from the combination of an indole molecule with a carboxyl group (Figure 6). In contrast, tryptophol contains a hydroxyl group, instead of a carboxyl bound to an indole, and is a regulator of quorum sensing in fungal colonies [24]. As such, both components are highly relevant markers for plant response to stress. However, their similar molecular structures make them difficult analytes to selectively detect. As such, they could not be easily determined by the incorporation of chromogenic or solvatochromic dyes alone.

Initially, to increase the sensitivity towards these components, one of the pH dyes (BG) was mixed with silver nitrate. Silver nitrate is a compound with a high redox potential, promoting reactions where silver can be reduced. On the contrary, indoles can be relatively easily oxidised, which could be exploited for selective detection. This oxidation reaction leads to a change in pH [25,26] that is reflected by an increase in measured signal. By exploiting this strategy, mixing BG and silver nitrate, the sensitivity of tryptophol detection increased by 153% towards tryptophol compared to the average signal. However, this strategy did not allow the detection of auxins, showing a signal 3% lower than the average. 

Cyclodextrin (CD) was tested as a potential specific compound to improve the selectivity towards both indole compounds tested in this study. CD is a cyclic oligosaccharide with an amphiphilic structure. This compound forms a pore, where the inner side is hydrophobic while the exterior is hydrophilic. A strong interaction between indoles and β-cyclodextrin has been demonstrated, showing promise for their incorporation to tailor the selectivity of colorimetric devices [27]. When CD was incorporated in the sensing mixture along with chromogenic dyes, the average signal response to indoles increased by 195%. An increase in signal was measured for both auxins and tryptophol. By combining CDs with silver nitrate, used previously used to enhance detection through redox-driven reactions, the sensitivity further increased by 258% (Figure 7a). As such, this combined strategy of specific molecular pores with silver nitrate was proven to be optimal to allow the detection of plant stress biomarkers.

#### 3.1.4. Exploiting Solvatochromic Properties of Reichardt’s Dye

As previously mentioned, besides chromogenic dyes, RD was also explored as a potential component in the detection of stress biomarkers. Contrary to BG, BR, and MR, RD is a solvatochromic dye, with colouration being influenced by the polarity of the environment. As such, its reactive properties change upon exposure to different solvents. This was exploited by drop-casting RD after being dissolved in different organic components: ethanol, methanol, acetone, as well as water. This component forms a solvation complex which influences the interaction with multiple gases. Chromatic differences in solvent complex formation could be directly observed during the deposition process. In general terms, the average sensitivity of the dyes towards each volatile decreased with the polarity of the solvent employed for the deposition (Figure 7b), reaching a minimum when water was used. Moreover, this dye showed a higher response to relatively large biomarkers (tryptophol and auxin, 2-hexen-1-ol) compared to simple, small molecules (acetone, ethanol, water, and methanol). Among all volatiles tested, 2-hexen-1-ol elicited the highest response, especially when acetone was used as the carrying solvent. However, a high response was also recorded in the presence of auxins, representing a potential interference in the testing. However, given the high relevance of both volatiles in plant abiotic stress, this dye was selected for production of the final sensor. 

#### 3.1.5. Selectivity Enhancement Using Graphene Quantum Dots

Although the combination of different dyes could lead to a measurable change in colour, the selectivity for certain analytes, such as 2-hexen-1-al, which do not contain an acidic or redox-active groups, was limited. Although some of the tested dyes, such as RD+Water, exhibited a high sensitivity towards this analyte, in the range of 20.58, the sensitivity when measured against other analytes such as auxin was also high, representing 40.7% of the signal obtained for 2-hexen-1-al. However, this dye combination showed a low selectivity, since it showed a high sensitivity towards multiple volatiles. This low selectivity hindered the accurate identification of gases using this dye alone. As such, a new signal amplification system, combining graphene quantum dots and chromogenic dyes was tested. Graphene quantum dots display a high photoluminescence due to their size and electrical properties [28]. Although these materials show a low toxicity, high stability, and photoluminescence, they have not yet been tested in optoelectronic noses. 

Despite its photoluminescent properties, the incorporation of graphene quantum dots did not increase the absolute signal intensity of the dyes. Both chromogenic dyes (BG, BR, and MR), as well as RD, experienced a decrease in sensitivity. This effect could be a consequence of the pH-dependent properties of graphene quantum dots [29], which interfered with the pH-driven reactions from chromogenic dyes. In the case of methyl red, for example, the highest measured signal after modification was achieved for acetone, ethanol, water, and acetic acid. On the contrary, methyl red without graphene quantum dots achieved the highest sensitivity in the case of tryptophol, acetic acid, and 2-hexen-1-al. These results suggest that the exchange of protons between the chromogenic dye and quantum dots could drive the sensitive response of these dyes. However, the lower sensitivity was a hindrance for their use as components in optoelectronic noses.

**Figure 7 micromachines-14-00314-f007:**
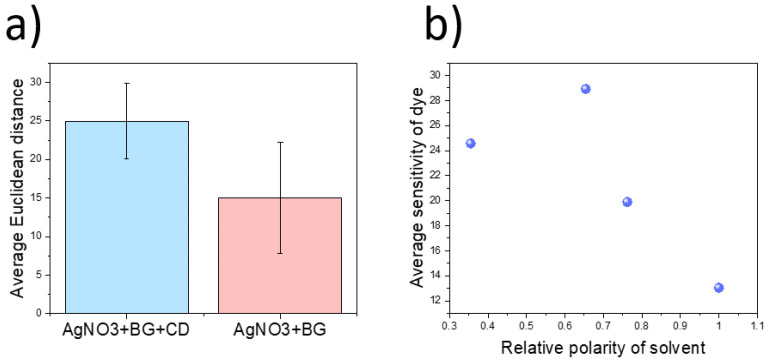
(**a**). Increase in sensitivity of detection of indole molecules using bromocresol green-based dyes upon the incorporation of cyclodextrin. The molecular pore within the cyclodextrin structure enabled an enhancement of indole detection. (**b**). Effect of relative solvent polarity on the dye sensitivity using RD. Values of relative solvent polarity have been extracted from [30].

#### 3.1.6. Final Selection of Dye Compositions

As described, an analysis of our screening results was conducted by calculating the difference in Euclidean distance of the dyes exposed to each gas compared to the average gas response across all tested volatiles. This test allowed us to determine which were the dyes that showed the highest absolute sensitivity towards each volatile. The dye combinations with the highest sensitivity values towards a certain volatile were considered good candidates for our final sensors. The final device combined dyes that showed the highest sensitivity to each studied volatile (acetone, ethanol, acetic acid, 2-hexen-1-ol, tryptophol, and auxin). However, since up to eight different dyes could be fitted inside each sensing paper, three different sensors that showed a high response towards 2-hexen-1-al were included, while only one selective dye was chosen for the other volatiles. A complete list of dyes and their respective selective volatiles is shown in Figure 8.

### 3.2. Sensing Device Calibration and Data Analysis

We have described the screening and selection of reactive dye sensors for simple volatiles, such as acetone, ethanol, and acetic acid, and plant-specific biomarkers, including tryptophol, auxin, and 2-hexen-1-al. In all cases, the sensitivity of the dyes was studied by subjecting the sensors to a concentration of 10 ppm of each dye and studying the signal change. However, it is also important to determine values for the limit of detection (LOD) and sensitivity of the devices. The determination of LOD in optoelectronic noses is key to ensure the applicability of the sensors in the determination of stress responses from plants, given the relatively low concentration of stress biomarkers within the initial stages of stress response. As such, the devices were initially calibrated in the presence of 2-hexen-1-al. This biomarker is involved in the response of plants to stress sources [31] and has been demonstrated to be relevant in the prediction of early abiotic stress [5]. As such, this biomarker represented our primary aim for detection, and three specific dyes were selected for its detection.

To determine the limit of detection of the devices, sensors were subjected to multiple concentrations of 2-hexen-1-al, ranging from 0.5 to 25 ppm. In each case, triplicate measurements were taken, and the average Euclidean signal ± standard deviation was plotted (Figure 9a). The linear range from our sensor was between 0.5 and 5 ppm, leading to one of the lowest limits of detections reported so far for colorimetric detection of volatile plant compounds. This detection limit was enough for the early detection of plant stress without the need for pre-concentration, common in the study of plant volatiles [32]. Moreover, our device could achieve a stable response within 1 h only, and each strip cost about GBP 1, including the price of the employed filter paper and the dyes, providing opportunities for application on large scale or in low-resource settings.

The use of eight different dyes could also be used for the specific identification of volatiles by principal component analysis (PCA). Using PCA, the 8-dimensional data (from the signal of the eight dyes), could be represented by a 3D plot, but retaining about 79% of the variability. PCA analysis is typically used in electronic noses where multiple gases are used to identify different volatile mixtures [33]. Within this method, the variability from different measurements can be represented in a single plot. The PCA plot revealed characteristic patterns for certain molecules, such as indoles, ketones, and carboxyl groups. In particular, 2-hexen-1-al measurements could be easily identified, since the measured values were allocated within a specific area (Figure 9b). These results show that a single dye was not sufficient to classify all gases. However, the collective signal differences from all dyes combined proved to be effective in profiling plant volatiles. 

Although the saturation point of the devices was relatively low (5 ppm), the devices could still provide qualitative evidence of the presence of stress within the early stages of exposure, which could be observed by the naked eye, which shows promise for the incorporation of this technology in the qualitative analysis of plant stress, without requiring electronics for measurement (Figure 9c). 

### 3.3. Assembly of Electronic Nose for Device Comparison

The results from this study were compared to those obtained by a custom-made electronic nose. This electronic nose contained typically used MQ metal oxide-based sensors, as well as infrared devices for CO_2_ detection. The use of this electronic nose enabled a comparison between the standard technologies used in the field with our low-cost optoelectronic nose. Similar devices have previously been employed in the non-invasive determination of soil moisture [34] or even for the diagnosis of COVID-19 [35]. In this case, devices with selectivity towards alcohols, O_3_, CO, and methane were employed. Moreover, infrared-based devices with selectivity were incorporated to allow a measurement of CO_2_. A total of seven sensors were added to the final sensing setup, able to measure eight different parameters (VOCs, temperature, humidity, and atmospheric pressure). A summary of the sensors employed and their selectivities is indicated in Table 1.

Firstly, the sensors were pre-conditioned for at least 48 h by initializing the devices, including the heating elements, until a stable signal was reached. Contrary to optoelectronic noses, which are readily available once they have been fabricated, the electronic nose requires time, in the range of hours to days, to be initialised and operative. These devices were connected to an Arduino microcontroller for the measurements, which could be used to automate the data collection process. After initialisation, the devices were assembled inside a septum-sealed box, connected to a tube as a gas inlet, and a gas outlet. Finally, to allow the determination of volatiles from plants, the Petri dishes were adapted by connecting a gas pump and a tube directed to the electronic nose. An initial testing of the system was conducted by incorporating a *Marchantia polymorpha* plant inside one of the Petri dishes. The differences between the measured values in open air and the Petri dish were then recorded and used to validate the applicability of this system for plant stress detection. 

### 3.4. Testing the Sensors In Vivo Using Marchantia Plants 

The sensor screening and characterisation process indicated a high performance of the final devices when eight dyes were employed. These sensors were tested in vivo using *Marchantia polymorpha* plants to determine the ability of the final devices to detect abiotic stress. Marchantia is a model plant widely used in research due to its rapid growth. A release of hexenal has also been reported in these plants, allowing the analysis of abiotic stress response through the detection of this component [36]. To induce a stress response, the plants were exposed to two common abiotic stressors: high salinity and nutrient starvation. High soil salinity is a major concern in agriculture, severely affecting more than six percent of the world’s agricultural lands [37]. Lack of drainage or improper irrigation are frequent causes of soil salinisation. Causes of nutrient starvation are often conditions that restrict photosynthetic activity and alter nutrient metabolism, and include drought, flooding, high and low temperatures, constant darkness, and defoliation. 

In this case, the sensors were tested with Marchantia plants over the course of 5 days by attaching the papers to the Petri dishes of growing plants for 1 h of each day (Figure 10e). In the case of high salinity, plants were subjected to GB medium supplemented with 50 mM of NaCl. Nutrient-deficient plants were produced by growth on GB medium that had been diluted by 1/100. The flexibility of our devices enabled the direct incorporation of the paper-based sensors inside Petri dishes without the need for further modifications. The volatiles produced by Marchantia plants were then analysed. By calculating the Euclidean distance of the colour change in our devices, the concentrations of 2-hexen-1-al could be estimated on all 5 days and compared across the three growth conditions (Figure 10a). Generally, 2-hexen-1-al concentrations decreased with time. During the first 3 days for the plants under high saline conditions, the trans-2-hexen-1-al values were higher than under control conditions. The same observation was made for the first two days in plants undergoing starvation. This is likely an effect of the stresses induced by the high salient and starvation conditions. A higher level of this volatile was observed in the case of NaCl overexposure compared to nutrient starvation. This observation is consistent with previously reported work on the abiotic stress response of plants, which indicate a release of hexenal molecules to enhance the protection of plants against high osmotic stress [38].

Although the use of our sensing device allowed an estimation of hexenal release by the plants due to abiotic stress, these measurements could not be easily used to diagnose early symptoms of stress in plants. Consequently, a PCA analysis was conducted using the measurements from the eight different dyes to better determine the specific patterns in plant responses (Figure 10b). By using PCA, the three plant populations subjected to each condition (control, high salinity, and nutrient deficiency) could be identified as early as day 2 of exposure to each abiotic stressor, since they were grouped in three distinctive areas. This allowed an early diagnosis of the signs of stress, even before plants showed any visible response, such as a change in leaf colouration or significant changes in size (Figure 10d). 

The results obtained from the optoelectronic nose devices were compared against those from the data obtained with an electronic nose. This e-nose comprises a combination of metal oxide-based sensors with multiple variable selectivities, targeting carbon monoxide, alcohols, methane, hydrogen, ozone/NOx, humidity, total amount of VOCs, and CO_2_. Similar to the previous case, a PCA analysis was conducted on the obtained results. By using PCA on the measurements of day 2, the high salinity- and starvation-stressed plants could not be differentiated from controls (Figure 10c). These results demonstrated a lower sensitivity of the electronic nose sensor compared to the optoelectronic nose. In addition, unlike our optoelectronic nose, the electronic nose could not be easily modified to show a higher selectivity towards specific analytes, such as indoles. Furthermore, Petri dishes had to be modified to enable the measurements, including an air pump and tube connection modifications. 

### 3.5. Spectrometer Design

The results obtained with the low-cost spectrometer showed a good performance in the identification and detection of the early signs of stress in plants, even allowing a detection of 2-hexen-1-al on the sub-ppm level. However, measurements showed variability in some cases due to some environmental factors, including background light intensity. To minimise the impact of the environment on the accuracy of detection while enabling automation of the data measurement process, a 3D printed spectrometer was designed. This spectrometer comprised a case with sliding lids, a rotor to move between each of the eight dye samples, and a Wio terminal (Figure 11a,b). The Wio terminal could display the numerical values of colour measurements from the optoelectronic noses in real time. Specifically, it showed the colour data in a graph and as numerical RGB values, as well as the HEX code of each colour (Figure 11c). This spectrometer could potentially be adapted to any number of dyes or sensor configurations, and could also be potentially interfaced with machine learning algorithms and wireless communication for IoT applications. The low-cost of this device combined with the easy operation could enable the use of this device by a wide community of farmers, horticulturists, and gardeners.

## 4. Conclusions

In this study, the selection of multiple chemo-responsive dyes and their application in an optoelectronic nose for non-invasive analysis of plant responses to stress has been demonstrated. Multiple dyes were screened to select those with optimal colorimetric responses to volatile biomarkers for abiotic stress, such as 2-hexen-1-al. Up to eight different dyes were examined for selective responses towards simple volatiles, such as acetic acid, acetone, and ethanol; indole molecules involved in the growth and fungal infections of plants, including auxins and tryptophol; and hexenal. A strong advantage of this approach is the flexibility of the devices, since the sensing dyes can be easily modified to change the selectivity towards different gases. Moreover, the optoelectronic nose developed as part of this work was non-invasive, it could be incorporated directly into Petri dishes without requiring modification, and it achieved a resolution high enough to monitor individual plants. This approach led to a significant improvement in cost compared to previously reported work, with an estimated cost of GBP 1 per strip, which is orders of magnitude lower than commercial e-noses. The simplicity of our device included ease of operability, only requiring a TCS34725 device for the data analysis. Moreover, some of the gases could be qualitatively evaluated by the naked eye given the changes in colour of the paper strips. By processing these measurements with a pattern classification algorithm such as PCA, the device enabled the identification of multiple volatiles, such as indoles and trans-2-hexen-1-al, showing a limit of detection on the sub-ppm level, one of the lowest values ever reported for an optoelectronic nose. 

Finally, our sensors were tested in vivo using *Marchantia polymorpha* as a model plant for abiotic stress response. Our optoelectronic nose was able to differentiate between plants undergoing abiotic stress from healthy plants as early as two days after exposure to the stressors, even prior to the manifestations of symptoms on the plant morphology or colouration.

One limitation of our measurement process is that the RGB colour sensor is only able to detect a homogeneous colour sample, not allowing the study of absorption gradients with the sensors. Moreover, the sensor papers cannot be reused, so for every measurement new sensors have to be manufactured. Further developments could include the optimisation of the sensor paper pore size and an automated classification of gases, for which a machine learning algorithm can be implemented. A simple algorithm such as the K-nearest neighbour (KNN) classification algorithm would be sufficient to achieve this goal. As such, this work reports for the first time a low-cost and highly sensitive paper-based optoelectronic nose for the early detection and monitoring of plant stress. This device has a plethora of applications in environmental analysis and agriculture, among others, and in particular, this approach could contribute to tackle food losses from wheat, corn, and rice, which are at high risk due to climate change.

## Figures and Tables

**Figure 1 micromachines-14-00314-f001:**
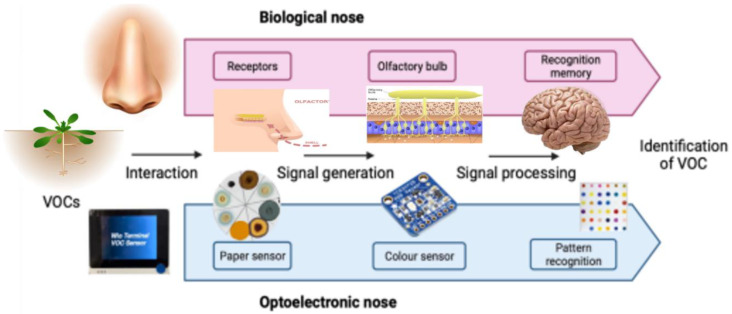
Schematic representation of the working principle for optoelectronic noses. These biomimetic devices use multiple cross-responsive sensor elements to produce a unique pattern of responses. Chemical properties that can be probed include, for example, proton acidity and basicity, redox potential, or presence of aminated groups. The combination of multiple colorimetric dyes leads to a distinct response to each analyte that allows the identification of analytes. In this work, a combination of chemo-responsive dyes was used to identify volatile organic compounds (VOCs) as a biomarker for plant stress.

**Figure 2 micromachines-14-00314-f002:**
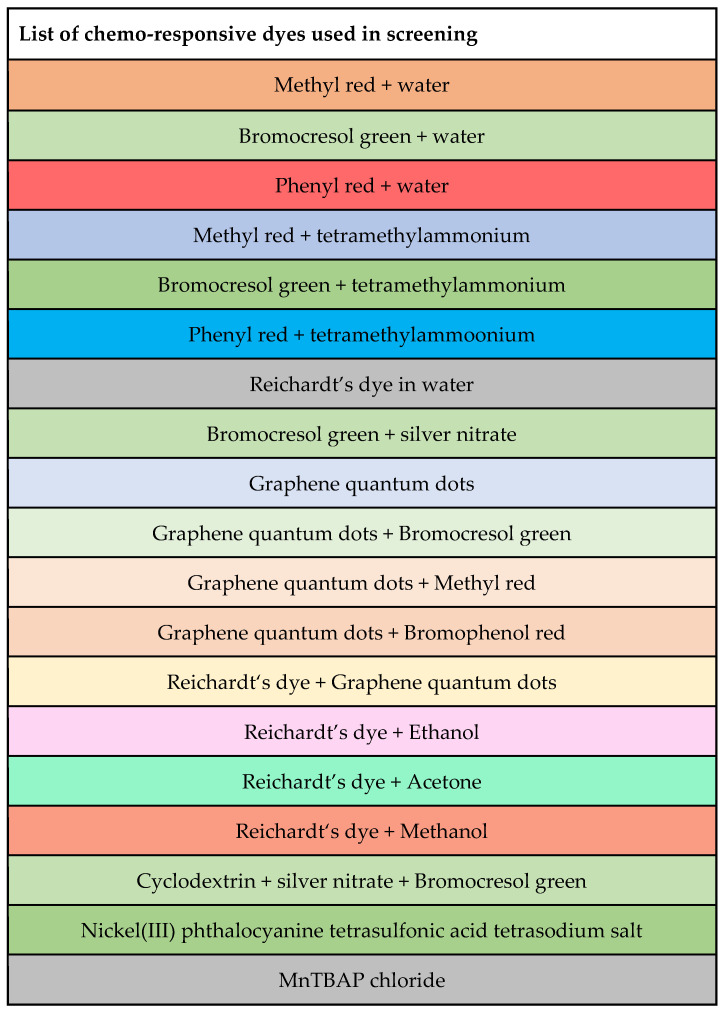
Composition of all dyes used during the screening process of the optoelectronic noses used in this study.

**Figure 3 micromachines-14-00314-f003:**
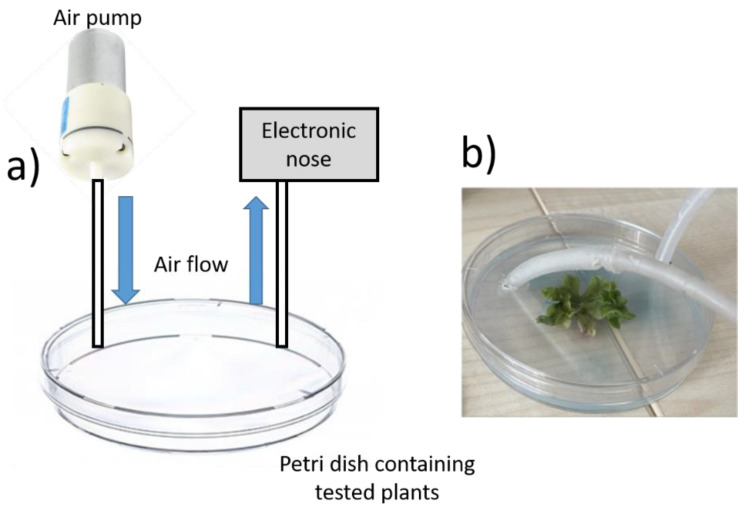
(**a**). Schematic representation of the setup used for sampling plant VOCs in Petri dishes with an electronic nose. Holes were inserted into Petri dish lids to access the interior air space, with tubes for connecting an air pump and to allow access to the electronic nose. An air flow was then established using an air pump to drive volatiles towards the electronic nose. (**b**). Real picture of the final setup, incorporating the tubes for the real-time measurement of plant volatiles.

**Figure 4 micromachines-14-00314-f004:**
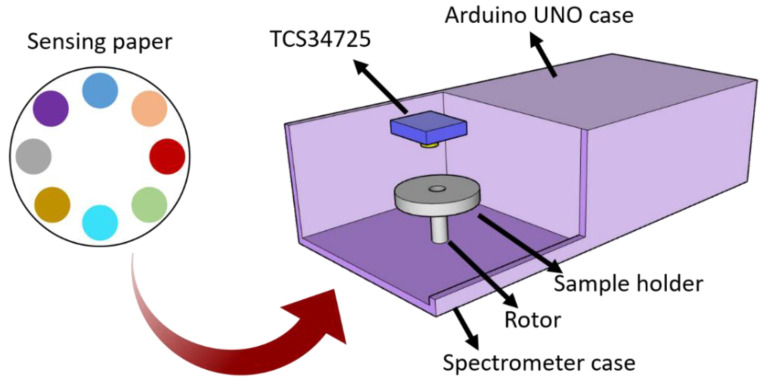
Schematic representation of the spectrometer device developed in this study for the determination of colorimetric changes in sensing papers. These papers, containing multiple chemo-responsive dyes, were characterised using a low-cost TCS34725 spectrometer, and the process was automated using a motorised rotor.

**Figure 5 micromachines-14-00314-f005:**
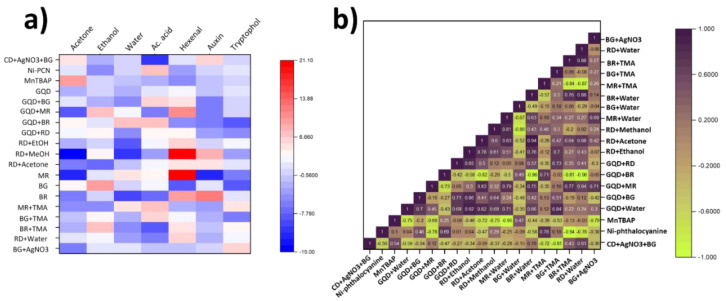
(**a**). Heatmap representation of the sensitivity of dye combinations towards each organic volatile studied in this work. (**b**). Correlation matrix between the performance of each dye against the studied gases. Darker colours indicate a higher correlation.

**Figure 6 micromachines-14-00314-f006:**
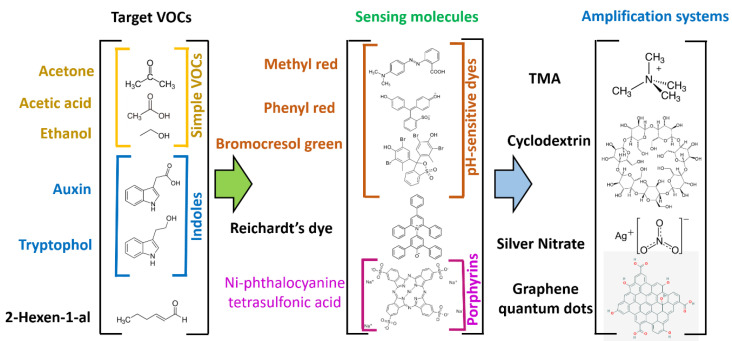
Schematic representation of compounds tested during the screening of dye compositions for volatile detection. The different dyes employed and signal amplification systems to enhance the selectivity are also indicated. These combinations of dyes allowed profiling of plant abiotic stress biomarkers.

**Figure 8 micromachines-14-00314-f008:**
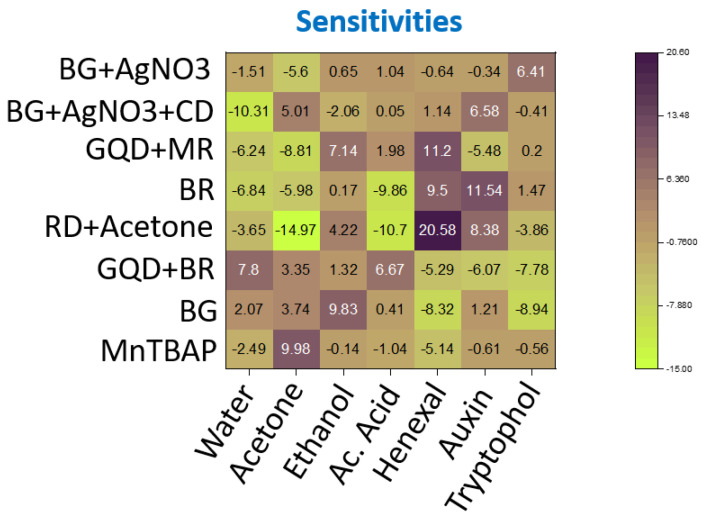
Heatmap of sensitivity results among the final selected dyes. Only one dye with high sensitivity towards each studied analytes (acetone, ethanol, ac. acid, auxin, and tryptophol), was selected. In the case of hexenal, three different dyes were selected.

**Figure 9 micromachines-14-00314-f009:**
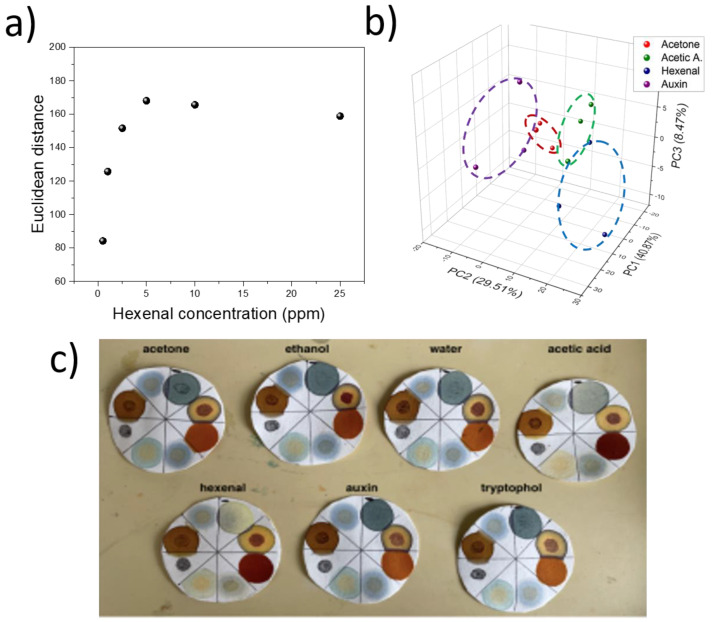
(**a**). Trans-2-hexen-1-al calibration plot using six different concentrations from 0.5–25 ppm. In each case, the average of the triplicate values is represented. (**b**). PCA analysis of the final device, containing eight dyes. The sensors were exposed to different volatiles and the first three principal components are represented. Each gas measurement has been highlighted with the characteristic value ranges. (**c**). Results of the same sensors being exposed to seven different gases (10 ppm) for 1h. The strips contained the final eight dye composition, and the name of the gas to which the respective sensor has been exposed is labelled above the sensor paper.

**Figure 10 micromachines-14-00314-f010:**
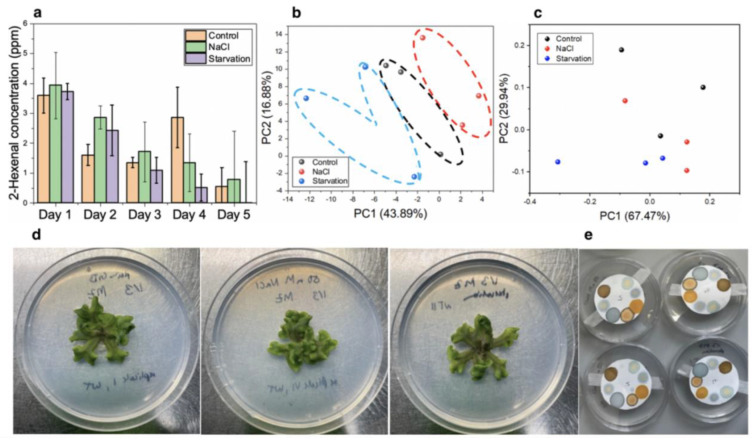
(**a**). Calculated trans-2-hexen-1-al concentrations in Marchantia over 5 consecutive days in three different growth conditions: high salinity, starvation, and control. Error bars represent the standard deviation of the triplicate measurements. (**b**). On day 2 of our measurements, the three different conditions could be differentiated using PCA. The results with the first two principal components in each case are shown. (**c**). On day 2 of our measurements, the three different conditions could not be differentiated using PCA and data from an electronic nose. (**d**). Pictures of *Marchantia polymorpha* on day 2 of the experiments in control, high salinity, and starvation media, from left to right, respectively. (**e**). The paper sensors were attached to the lid of the Petri dish for 1h to react with the plant VOCs before measurements were taken.

**Figure 11 micromachines-14-00314-f011:**
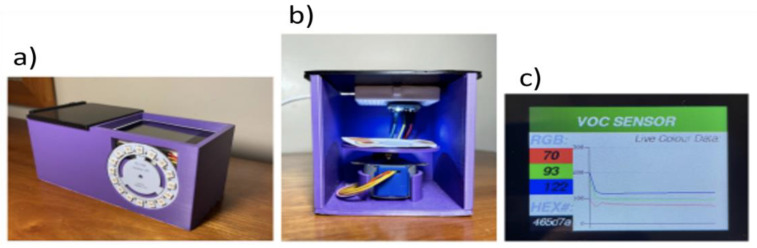
(**a**,**b**). Spectrometer design, comprising a case with sliding lids, a rotor to move the dye samples, and a display to show live colour results. A ring-shaped LED indicates the measurement status of the spectrometer. During the measurement process, the optoelectronic nose and TCS34725 spectrometer were kept in the dark. However, the samples were illuminated by a white LED on the RGB sensor. (**c**). The Wio terminal display showing real time colour data in a graph and as numerical RGB values, as well as the HEX code of each colour.

**Table 1 micromachines-14-00314-t001:** Summary of sensors incorporated in our electronic nose for results comparison with an optoelectronic nose.

Sensor	Parameters
SCD30	CO_2_, temperature, humidity
BME680	tVOC, temperature, humidity, atmospheric pressure
MQ2	Butane, methane, alcohol, hydrogen
MQ3	Alcohols, ethanol
MQ4	Methane
MQ7	Carbon monoxide
MQ131	Ozone, NO_x_

## Data Availability

Not applicable.

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
