# Peer review of "Paper-Based Multiplex Sensors for the Optical Detection of Plant Stress"

_micromachines, 2023, doi:10.3390/mi14020314_

Round 1
Reviewer 1 Report
Dear authors
This is an interesting research related to development of paper-based multiplex sensors for early diagnosis of plant diseases. I think this concept is novel. Moreover, this manuscript is also well-written and organized. However, it should be revised some following issues before accepting for publication
1. Please check all abbriviations in this manuscript. The authors should provide a full expression for the first appearence. For example, page 2, CNT, IR, RGB
2. Equations should be numbered (Page 6)
3. In the section 2, sub-section "2.1. Materials" should be inserted
4. Table 1 should be changed to Figure because they have colors
5. Figure 9 is very poor. Please provide another one with high resolution
6. "Conclusion" is too long. it should be significantly revised regarding emphasize the important results and limitations of this research. Moreover, some future approaches can be pointed
Author Response
We would like to thank the reviewer for the positive comments on our manuscripts, and especially for indicating that our research is innovative and interesting. Below is a reply to the revisions:
- Please check all abbreviations in this manuscript. The authors should provide a full expression for the first appearence. For example, page 2, CNT, IR, RGB
Acronyms and abbreviations have been corrected throughout the manuscript.
- Equations should be numbered (Page 6)
Equations have been numbered, and references within the main text.
- In the section 2, sub-section "2.1. Materials" should be inserted
Sub-section 2.1. has been inserted (P4).
- Table 1 should be changed to Figure because they have colors
Table 1 has been changed to “Figure 2” (P5), and figure numbers have been updated accordingly throughout the text.
- Figure 9 is very poor. Please provide another one with high resolution
Figure 9 (now figure 10, P19), has been replaced by a higher quality picture.
- "Conclusion" is too long. it should be significantly revised regarding emphasize the important results and limitations of this research. Moreover, some future approaches can be pointed
Conclusions have been shortened, and limitations and future potential applications have been included (P20).
Reviewer 2 Report
In the paper “Paper-based multiplex sensors for the optical detection of plant stress” – Micromachines 2166957, the authors present a low-cost solution for early detection and monitoring of plant stress based on a highly sensitive paper-based optoelectronic nose.
The introduction is clear, justified with appropriate references, and relevant to frame the work developed. Figure 1 is an excellent “graphical abstract” that lets readers quickly understand the work’s purpose. Experimental sections were well structured, and results were well presented and discussed.
The work was well conducted and discussed, and I found everything clear and easy to understand. Therefore, I recommend the article be accepted in Micromachines, but I have a question for the authors related to room temperature and its influence on VOCs emission.
Temperature increase accelerates the biosynthesis of VOCs and their diffusion from plant tissues. Besides, room temperature could change several degrees during a workday. Did the authors take any steps to control the sampling temperature? How could room temperature affect the electronic nose results? A small comment respecting this subject can contribute to improving the manuscript.
Below, I list some minor corrections that authors should consider to improve the manuscript.
Line 164 - “The Materials” should be section “2.1 The materials” with the respective correction of the following sections.
Line 186 - For consistency of the manuscript, please correct, “table 1” should be “Table 1”
Line 253 - Please correct: “Li et al. “ should have “et al.” in italics
Line 276 - Please, correct: “trypotophol)“ to tryptophol and eliminate the bracket “)”
Line 459 - Please, correct: “tetram-“ to “tetra-“
Line 535 - Please, correct: “fpr” to “for”
Line 671 - Please, correct: “CO2” to “CO2” with the number 2 in subscript
Line 679 - Please correct: “table 2” should be “Table 2”
Line 806 - Please correct: “tcs34725” to “TCS34725”
Please review the references. Some examples of missing data are listed below:
Ref 3, 22 - References should be more complete. A valid URL should be included if the reference is from a website.
Ref 4, 7, 8, 19, 24, 30, 32, .. - The journal name is missing.
Although these minor corrections, reading and revising your work was a pleasure. Well done!
Author Response
We would like to thank the reviewer for the positive comments on our manuscripts, and indicating that the manuscript was easy to follow. Below is a reply to the revisions:
Temperature increase accelerates the biosynthesis of VOCs and their diffusion from plant tissues. Besides, room temperature could change several degrees during a workday. Did the authors take any steps to control the sampling temperature? How could room temperature affect the electronic nose results? A small comment respecting this subject can contribute to improving the manuscript.
We are aware of the changes in volatile production due to environmental conditions. This is why plants were grown and tested under controlled conditions, and a constant temperature of 21o, which is the standard temperature for growing Marchantia. As such we do not foresee any acceleration effect on biosynthesis and/or VOC diffusion as a result. This has now been included within the “Materials and Methods” section.
Line 164 - “The Materials” should be section “2.1 The materials” with the respective correction of the following sections.
The title has been updated as requested.
Line 186 - For consistency of the manuscript, please correct, “table 1” should be “Table 1”
Table 1 has been renamed as “Figure 1” as requested by Reviewer 1.
Line 253 - Please correct: “Li et al. “ should have “et al.” in italics
“Et al.” has been italicized
Line 276 - Please, correct: “trypotophol)“ to tryptophol and eliminate the bracket “)”
This has been now corrected.
Line 459 - Please, correct: “tetram-“ to “tetra-“
This has been corrected
Line 535 - Please, correct: “fpr” to “for”
This has been corrected
Line 671 - Please, correct: “CO2” to “CO2” with the number 2 in subscript
This has been corrected
Line 679 - Please correct: “table 2” should be “Table 2”
Table 2 (now table 1) has been amended accordingly
Line 806 - Please correct: “tcs34725” to “TCS34725”
This has been corrected
Please review the references. Some examples of missing data are listed below:
Ref 3, 22 - References should be more complete. A valid URL should be included if the reference is from a website.
On reference 3, the website has been included. Reference 22 is from a conference proceeding, which has been specified.
Ref 4, 7, 8, 19, 24, 30, 32, .. - The journal name is missing.
Journal names have been added on the references.
Round 2
Reviewer 1 Report
It can be accepted for publication in the current form